# Cost effectiveness of empagliflozin in adult patients with chronic kidney disease in the Netherlands

**Bart Slob** [1,2] *, **Tanja Fens** [1,2], **Maaike Weersma**[3], **Maarten Postma**[1,2,4,5,6], **Cornelis Boersma**[1,2,7], **Lisa de Jong**[1,2]

1 Department of Health Sciences, University Medical Center Groningen, University of Groningen, Groningen, The Netherlands, 2 Health-Ecore Ltd, Groningen/ Zeist, The Netherlands, 3 Boehringer Ingelheim bv, Amsterdam, The Netherlands, 4 Department of Economics, Econometrics & Finance, Faculty of Economics & Business, University of Groningen, Groningen, The Netherlands, 5 Department of Pharmacology and Therapy, Faculty of Medicine, Universitas Airlangga, Surabaya, East Java, Indonesia, 6 Center of Excellence in Higher Education for Pharmaceutical Care Innovation, Universitas Padjadjaran, Sumedang, West Java, Indonesia, 7 Department of Management Sciences, Open University, Heerlen, The Netherlands

☯ These authors contributed equally to this work.
* bartslob@health-ecore.com

**Data Availability Statement:** All relevant data are included in the manuscript. The analyses were conducted based on publicly available information

## Abstract

### Aim

The recent EMPA-KIDNEY trial showed evidence for preventing disease progression in adult patients with chronic kidney disease (CKD) treated with empagliflozin. It is however yet unknown if use of empagliflozin is cost effective in the Netherlands. We aimed to evaluate the cost effectiveness of empagliflozin in adult patients with CKD in the Netherlands.

### Methods

A cost-effectiveness analysis was conducted using a Markov state microsimulation model, simulating kidney progression of CKD patients with eGFR <90 ml/min per 1.73 m2 comparing empagliflozin plus standard of care (SoC) and SoC alone. KDIGO classification was used to describe the risk of CKD progression. The input data were taken from the EMPA-KIDNEY trial (baseline characteristics, treatment effect, and utilities), and published data and national sources were used for general population mortality, treatment and event costs. The analyses were performed from a societal perspective with applying a lifetime horizon. Discounting was done according to the Dutch pharmacoeconomic guidelines. The incremental cost-effectiveness ratio (ICER) was compared to a willingness-to-pay threshold of €50,000/QALY. Deterministic and probabilistic sensitivity analyses were performed to explore the impact of uncertainty around the input parameters.

### Results

The base-case results showed total discounted costs for empagliflozin plus SoC and SoC alone of €200,193 and €234,574 respectively, indicating total savings of €34,380. Empagliflozin plus SoC was associated with higher total discounted health benefits of 11.06 life

which is presented and referenced in the manuscript and its supplementary materials.

**Funding:** The study was supported and funded by Boehringer Ingelheim (BI). BI was given the opportunity to review the manuscript for medical and scientific accuracy as well as intellectual property considerations. The funding organization did not play a role in the study design, data collection and analysis, decision to publish, or preparation of the manuscript and only provided financial support in the form of authors' salaries and a research fund to conduct the analysis. BI provided support in the form of salary for author MW and Health-Ecore received a research fund to conduct the analysis and provided support in the form of salary for authors BS, TF, LJ and CB and MJ are shareholders of this company, but this did not have any additional role in the study design, data collection and analysis, decision to publish, or preparation of the manuscript. The specific roles of these authors are articulated in the 'author contribution section.

**Competing interests:** CB and MP receive grants and honoraria from various pharmaceutical companies, including Boehringer Ingelheim. They are both shareholders of Health-Ecore, the Netherlands. TF, BS and LJ are employed as consultants at Health-Ecore, which received a consultancy fee for the conduct of this study. MW is an employee at Boehringer Ingelheim Netherlands, the funder of this study. The commercial affiliations of the authors do not alter our adherence to PLOS ONE policies on sharing data and materials. The EMPA-KIDNEY trial was initiated, designed, and conducted by the University of Oxford in collaboration with a Steering Committee of experts and Boehringer Ingelheim. The presented analyses were initiated and conducted by Boehringer Ingelheim independently from the EMPA KIDNEY Collaborative Group. The authors meet criteria for authorship as recommended by the International Committee of Medical Journal Editors (ICMJE). The authors did not receive any personal payment related to the development of this manuscript. Boehringer Ingelheim was given the opportunity to review the manuscript for medical and scientific accuracy as well as intellectual property considerations.

years (LYs) and 9.01 quality-adjusted life years (QALYs), compared with 9.74 LYs and 7.79 QALYs for SoC alone, resulting in an additional 1.31 LYs and 1.22 QALYs for empagliflozin plus SoC. Empagliflozin plus SoC is a dominant alternative compared to SoC alone. Sensitivity analyses confirmed the robustness of the findings and conclusion.

## Conclusion

Using empagliflozin in addition to SoC in adult patients with CKD is likely to be cost saving compared to the current SoC in the Netherlands, irrespective of diabetes status and albuminuria.

## Introduction

Chronic kidney disease (CKD) is a progressive condition with reduced renal function and/or increased albuminuria or specific abnormalities in kidney structure or function, lasting more than three months [1]. The disease is associated with various health consequences and complications such as cardiovascular, metabolic or infectious diseases, with incidences increasing proportionally with CKD progression [2–4]. This may ultimately lead to kidney failure and death [5].

CKD presents a significant burden to the healthcare system. A global CKD prevalence of 9.1% was reported for 2017, with an expected increasing trend for the coming decades [6]. Particularly people with low income, low education and older age (above 60-years-old) are at risk of developing CKD with a prevalence between 27.6% and 34.3% [6, 7]. In the Netherlands, CKD prevalence is estimated to be around 12% [8]. However, a retrospective study showed that only 5.1% of the population is diagnosed [9]. The relatively high prevalence and low diagnosis numbers emphasize the need to improve preventive measures that can slow the progression of the disease and reduce the resulting harm (morbidity and mortality).

Disease management strategies are dependent on the disease stage, mostly defined by the estimated glomerular filtration rate (eGFR) and urine albumin-creatinine ration (uACR), while the Kidney Disease Improving Global Outcomes (KDIGO) classification is mostly used to describe the risk of CKD progression [1]. CKD management involves treatment strategies to address the root cause of CKD and comorbidities, to prevent disease progression to end-stage kidney disease (ESKD), which is defined as an eGFR below 15 mL/ min/ 1.73m2 or the need of kidney replacement therapy (KRT) in the form of dialysis or transplantation [1]. Medical treatment of CKD consists of cardiovascular risk management with statins (possibly in combination with ezetimibe) and blood pressure lowering agents. For patients with moderately to highly albuminuria, treatment with renin-angiotensin system (RAS) inhibitors is recommended. Non-medical strategies include lifestyle and dietary adjustments for improving cardiovascular health, or comorbidity effects of hypertension or diabetes [1].

In addition to the medical complexity of CKD management, there is a high hospitalization burden that considerably impacts the health care budget [10]. CKD costs are estimated to match those of cancer and diabetes, and on a European level they reach up to 140 billion euros [11]. The highest burden in terms of costs, mortality and morbidity is caused by patients in ESKD. Patients on dialysis experience a low quality of life, thereby incurring costs ranging from €80,000 to €120,000 per patient per year in the Netherlands [12]. Kidney transplantation can considerably improve the quality of life and survival of patients in ESKD; however, this also comes at a cost of approximately, €80,000 per transplant [12].

Europe has identified the need for a systematic approach to fight CKD, including awareness, prevention, early detection and optimal care [11, 13]. The evidence for applying preventive medical management for CKD progression by adding sodium-glucose cotransporter 2 (SGLT2) inhibitors to the current standard of care (SoC) shows potential for decreasing the clinical and economic burden on the healthcare system.

The update of the KDIGO 2024 Clinical Practice Guideline for the Evaluation and Management of Chronic Kidney Disease issued a recommendation to treat patients with Type 2 Diabetes (T2D), CKD and an eGFR >20ml/min/1.73m$^2$ with an SGTL2 inhibitor. In addition, SGLT2 inhibitors are recommended for treating adults with CKD with eGFR ≥20 ml/min/1.73m2 and urine ACR ≥200 mg/g, or heart failure, irrespective of level of albuminuria [1].

SGLT2 inhibitors are also recommended for adult patients with CKD with eGFR 20 to 45 ml/min/1.73m$^2$ with uACR<200 mg/g. In the Netherlands, SGLT2 inhibitors are the preferred choice of therapy in T2D patients with a very high risk for cardiovascular events due to its shown benefits in preventing cardiovascular disease (CVD) events and progression of CKD [14]. For patients with CKD without T2DM, the recommendation is to initiate treatment with SGLT2 inhibitors in addition to maximum tolerated renin-angiotensin-aldosterone system (RAAS) blockade in patients with very high KDIGO risk and an eGFR >20ml/min/1.73m$^2$ [15].

Empagliflozin is a potent, selective, reversible SGLT2 inhibitor, recently approved for the treatment of adult patients with CKD [16]. During two years of follow-up in the EMPA-KIDNEY trial, empagliflozin showed to significantly reduce the risk of kidney disease progression or death from cardiovascular causes compared to placebo among a wide range of patients with CKD who were at risk for disease progression [17]. The EMPA-KIDNEY trial examined a broad albuminuria spectrum to include patients who were not included in previous CKD studies, or who were significantly underrepresented compared to the actual CKD population. Based on these positive results, the Dutch National Health Care Institute (*Zorginstituut Nederland*) has given positive advise for the reimbursement of empagliflozin for the treatment of CKD [18]. Cost effectiveness was not part of this assessment as another SGLT2 inhibitor was already reimbursed for the CKD indication and therefore concerned a 1A application for which cost effectiveness is not required [19].

The KDIGO guideline highlights the need for economic evaluations, especially in people with CKD without diabetes and low levels of albuminuria to establish their level of cost-effectiveness. Reducing cost burden of hospitalizations and dialysis is highly desirable, and quality of life may be preserved longer from their avoidance. Specifics as to the costs of these medications offset the health gains will be country-dependent. Such country-specific analyses are necessary, as healthcare systems, guidelines, WTP-thresholds, and methods for calculating cost-effectiveness vary across countries. By focusing on the Netherlands, this study will highlight the value of empagliflozin within the Dutch healthcare context. The findings can help policymakers identify where its added value lies. Currently, the cost effectiveness of empagliflozin for the treatment of CKD in the Netherlands is unknown. Therefore, we aimed to evaluate the cost effectiveness of empagliflozin in adult patients with CKD in the Netherlands, irrespective of diabetes status and albuminuria

## Methods

A cost-effectiveness model was developed to evaluate the cost effectiveness of empagliflozin plus SoC compared to SoC alone in adult patients with CKD from a Dutch societal perspective. The analysis was conducted in accordance with the Dutch guideline for economic evaluations in healthcare [20] and reported in line with the Consolidated Health Economic Evaluation Reporting Standards 2022 (CHEERS 2022) checklist [21].

## Model structure

A Markov state microsimulation model was developed in Microsoft Excel®. Microsimulation was used because it allows to simulate individual patients by randomly distributing baseline characteristics within specific limits and to track individual disease histories over time, as well as to study the impact of these factors. This is necessary for a heterogeneous population like patients with CKD. To reflect the CKD progression, the model's health states were defined as per the KDIGO classification, which is based on eGFR (KDIGO eGFR category G2-G5) and uACR (KDIGO uACR category A1-A3), with a higher stage indicating a higher risk for disease progression [1]. Health state G1 (eGFR > 90 ml/min/1.73m$^2$) was not included in the model since this is not considered CKD according to KDIGO criteria. The distribution over the different health states was calculated by multiplying the proportion of the patients in the EMPA-Kidney trial in each eGFR category (G2-G5) by the proportion in each uACR category (A1-A3). In all health states, the patient can experience the same set of complications, including ESKD, cardiovascular disease, diabetes, hypertension, cancer, bone and mineral disorders, infections, acute kidney injury (AKI), anemia, which were modelled in separate sub-models (Fig 1). However, the risk of these complications was health state-specific and varied based on eGFR, uACR, and other factors such as glycated haemoglobin (HbA1C) and body mass index (BMI). Annual cycles were assumed using half-cycle correction. Details on the sub-models and the model engine of the natural disease progression can be found in the S1 Appendix.

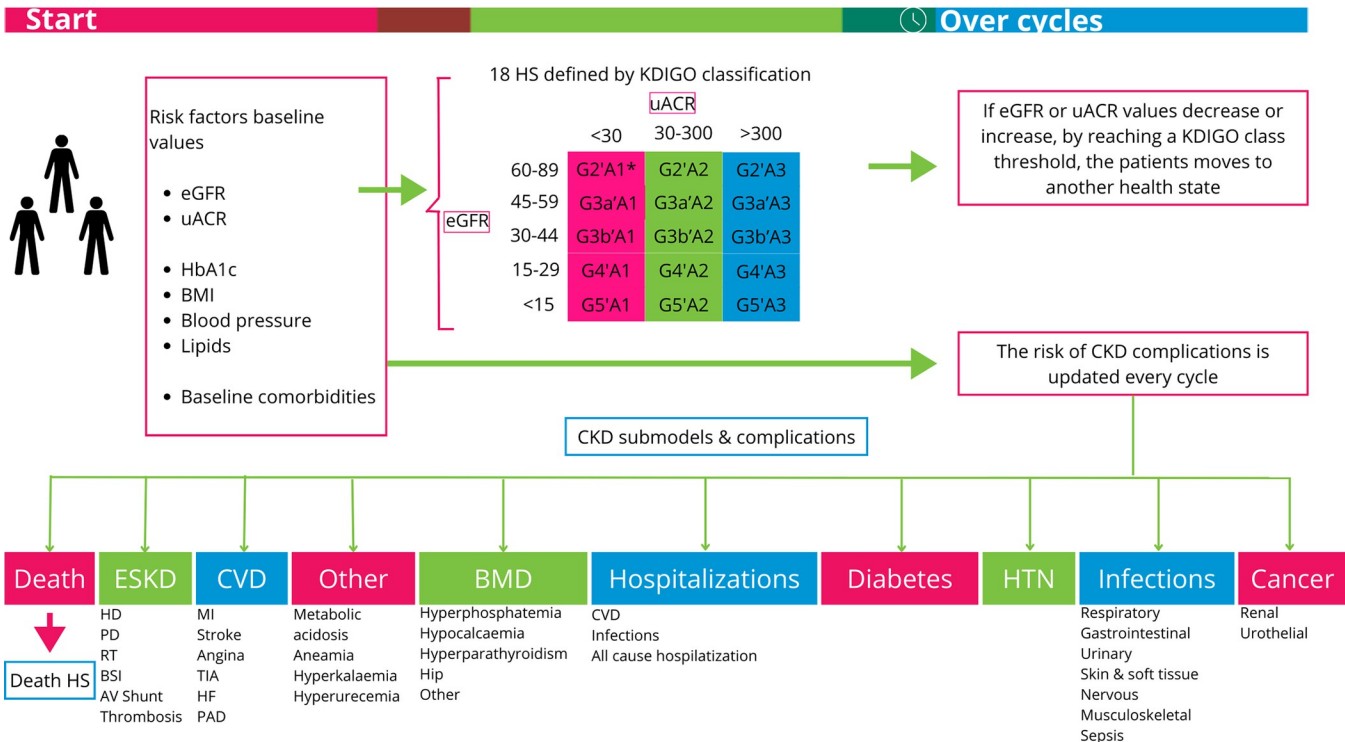

**Fig 1. Health states, complications and events included in the CKD disease progression model.** Abbreviations: A = KDIGO uACR category (in mg/g), AV = arteriovenous, BMD = bone and mineral disease, BMI = body mass index, BSI = blood stream infection, CKD = chronic kidney disease, CVD = cardiovascular disease, eGFR = estimated glomerular filtration rate, ESKD = end-stage kidney disease, G = KDIGO eGFR category (in mL/min/1.73$^2$), HD = hemodialysis, HF = heart failure, HTN = hypertension, HS = health state, KDIGO = Kidney Disease: Improving Global Outcomes, MI = myocardial infarction, PAD = peripheral arterial disease, PD = peritoneal dialysis, RT = renal transplantation, TIA = transient ischemic attack, uACR = urine albumin-creatinine ratio. *Even though this is not considered as CKD according to KDIGO criteria, an assumption has been made to include this in the model to model the disease progression, however, treatment effects were not considered in this group.

### Perspective, time horizon, and discounting

In the base case, the incremental cost-effectiveness ratio (ICER) was estimated from a societal perspective with a life-time horizon (up to 100 years). Discount rates applied were 4% for costs and 1.5% for effects [20].

### Baseline characteristics and treatment effect

The baseline characteristics were based on the EMPA-KIDNEY trial [17], with an average age of 63.8 years and 66.8% being men (S1 Table). At baseline, the distribution of eGFR and uACR levels were based on the levels of the included patients in the EMPA-Kidney trial (S1 Table). The eGFR and uACR levels of included patients were used in a matrix to estimate the baseline distribution of health states. Due to the inclusion criteria of the trial, no patients started in the G5 health state at baseline. However, it is possible for patients to transition to this health state in the model at a later stage. Treatment effects were incorporated in the model as annual changes in eGFR and uACR per health state, based on the findings of the EMPA-KIDNEY trial [17]. No treatment effect was included for health states G2A1, G3aA1, and G5A1-3 as these populations were not included in the EMPA-Kidney trial. Hazard ratios on event rates for hospitalized heart failure and AKI were additionally taken into account [17]. The duration of the treatment effect on effects and costs was assumed to last over the entire time horizon, except after treatment discontinuation.

### Mortality

The risk of death was considered in three ways: non-specific mortality, CVD death, and renal death. Non-specific mortality was based on Dutch national life tables after excluding mortality from CVD and renal diseases as causes of death [22]. The risk of CVD death was based on mortality risk engines from the CKD-PC registry specific in the CKD population including data from 41 countries [23]. Renal death accounts for all fatal events occurring after patients have initiated KRT and was based on the UKRR annual report [24].

### Adverse events

Adverse event rates per 100 patient-years were applied in the model. Amongst these, lower limb amputations have been considered for which data has been sourced from the EMPA-KIDNEY trial findings [17].

### Utilities

The health state utilities were based on EQ-5D-5L data collected in the EMPA-KIDNEY trial [17]. For the comorbidities and complications modelled in the sub-modules, disutilities from literature were used [25, 26]. All utilities were recalculated to reflect Dutch values using a conversion of UK to Dutch utilities formula [19, 27]. All (dis)utilities are summarized in S2 Table.

### Costs

We compared the effects of empagliflozin 10mg once daily, in addition to SoC, compared to SoC alone. SoC costs were calculated as weighted annual mean, based on the concomitant medication use of renin-angiotensin-system inhibitors (angiotensin-converting enzyme inhibitors and angiotensin receptor blockers) in the EMPA-KIDNEY trial [17]. Drug prices were based on official list prices including 9% VAT [28]. Costs involved in the management of CKD and the associated complications and events modelled were tracked as CKD progresses over time. The costs were assigned for the specific health states, complications, and events in each

cycle of the model and were classified in different categories: 1) Health state costs consisting of monitoring costs associated with the health states; 2) Event costs: first year hospitalization costs upon occurrence of an acute event; 3) 'Follow-up' costs of complications after the first year, and 4) Adverse event costs (e.g. for leg amputation). These costs were sourced from (national) literature. Given the societal perspective, direct costs outside the healthcare sector and indirect costs were also considered in our model. These costs were only considered for the health states and ESKD-related complications (dialysis and KRT). The cost parameters are presented in S3 Table. All costs were inflated to 2022 euros using inflation rates from Dutch national statistics [22].

## Model outcomes

The main model outcomes were the ICER and the net monetary benefit (NMB), which allow to determine whether the intervention is cost effective or not. According to the Dutch pharmacoeconomic guidelines, the ICER was compared to a willingness-to-pay (WTP) threshold of €50,000/QALY based on the disease burden corresponding to a proportional shortfall of 0.43 [29]. Life expectancy, incidence of hospitalizations and events, time to ESKD, and mortality were also presented, as well as the ICER per life year (LY) gained. Additionally, a cost breakdown was presented by category.

## Sensitivity and scenario analysis

Sensitivity analyses were performed to explore the impact of uncertainty around the input parameters. A one-way sensitivity analysis was done by varying each single input sequentially while holding other parameters fixed and assessing the effect on the NMB. This gave an indication of the most influential parameters of the model. A probabilistic sensitivity analysis (PSA), in which all parameters (i.e., ratios, probabilities, utilities, costs) are varied simultaneously over their uncertainty ranges, was also conducted. The input parameters for mortality, the treatment effect and (dis)utilities were sampled from a beta distribution, cost parameters from gamma, and treatment effects (Hazard Ratios) from log-normal distributions. A total of 500 simulations were conducted to test the robustness of the model results.

Several scenario analyses were conducted: 1) healthcare payer's perspective, 2) EMPA-KIDNEY subgroup with diabetes [17], 3) EMPA-KIDNEY subgroup without diabetes [17], 4) patient baseline characteristics based on Dutch data from Sundström et al. [30] (only concerning age and gender, while the other characteristics remained the same as in the base-case analysis), 5) health state utilities based on Jesky et al. calculated to Dutch values using the same formula used in the base case [31], 6) a time horizon of 10 years, 7) a time horizon duration of 20 years, and 8) including all the SoC costs for the comorbidities (as a weighted average) as reported in the EMPA-Kidney trial.

## Ethics statement

The analyses were conducted based on publicly available information which is presented and referenced in the article and Supplementary Material files, and did therefore not require any patient consent forms or approval from an ethical review board. Furthermore, some of the input data was obtained from the EMPA-Kidney trial which was approved by ethics committees at each center, with all patients having provided written informed consent (full information can be found in the original publication of this trial [17]). All patient data used was anonymous.

## Results

### Base case analysis

Treatment with empagliflozin plus SoC resulted in a higher life expectancy compared to the treatment with SoC alone (11.06 vs. 9.74 LYs). Furthermore, simulated patients treated with empagliflozin plus SoC showed a diminished incidence rate per 100 patient-years across various health outcomes compared to SoC, including all cause hospitalizations (30.78 vs. 31.55), hospitalizations related to CVD (6.99 vs. 7.28), AKI events (3.33 vs. 3.79), HF hospitalizations (2.59 vs. 2.89), initial hospitalizations due to infections (20.06 vs. 20.29), and mortality (7.84 vs. 9.02). Moreover, treatment with empagliflozin plus SoC demonstrated an extended time to ESKD, defined either as a decline an eGFR below 15 ml/min/1.73 m$^2$, or the initiation of KRT compared to SoC alone. Specifically, the observed temporal difference was 10.79 years in the patients treated with empagliflozin and SoC versus 7.82 years in patients treated with SoC alone. For a full overview of the incidence of events as per sub module for the patients treated with empagliflozin plus SoC versus SoC see S4 Table. The reduced occurrence of clinical events among simulated patients receiving empagliflozin plus SoC in comparison to those solely receiving SoC led to lifetime incremental discounted gains of 1.31 LYs and 1.22 QALYs. This outcome translated into a dominant ICER and a NMB of €95,183 (Table 1).

Patients receiving empagliflozin plus SoC incurred higher lifetime treatment and monitoring costs. The costs for most complications were also slightly higher for empagliflozin plus SoC compared to SoC alone, which is caused by the extended life expectancy. On the other hand, empagliflozin plus SoC incurred considerably lower costs associated with KRT and conservative therapy during ESKD, resulting in a net cost saving of €34,380 per patient (Table 2).

### Sensitivity analysis

The one-way sensitivity analysis revealed that the most influential input parameters on the NMB predominantly concerned the incremental treatment effect associated with the G4*A2 health state for empagliflozin plus SoC, particularly concerning risk factor progression for uACR and eGFR. Other influential parameters included the costs associated with acute events, especially those for hemodialysis for patients with ESKD, and the increment risk factor progression for uACR in the G4*A3 health state in both the empagliflozin plus SoC as well as the SoC alone arm (Fig 2).

The outcomes of the PSA underscored the robustness of the findings, positioning 99.8% of the results within the dominant quadrant of the cost-effectiveness plane (Fig 3). Furthermore, the cost effectiveness acceptability curve revealed that the probability of empagliflozin plus SoC being cost effective is 100% at a WTP-threshold of €50,000/QALY (S1 Fig).

**Table 1. Deterministic results of the base-case analysis.**

| Outcome | Empagliflozin plus SoC | SoC | Incremental |
|---|---|---|---|
| Total discounted costs (€) | 200,193 | 234,574 | -34,380 |
| Total discounted LYs | 11.06 | 9.74 | 1.31 |
| Total discounted QALYs | 9.01 | 7.79 | 1.22 |
| Incremental cost (€) per LY | Dominant (lower costs, more LYs) | | |
| Incremental cost (€) per QALY | Dominant (lower costs, more QALYs) | | |
| NMB (€) | 95,183 | | |

Abbreviations: LY = life year, NMB = net monetary benefit; QALY = quality-adjusted life year, SoC = standard of care

**Table 2. Breakdown of costs outcomes of the base-case analysis.**

| Cost category (€) | Empagliflozin plus SoC | SoC | Incremental |
|---|---|---|---|
| Monitoring | 41,514 | 38,795 | 2,719 |
| Treatment | 2,737 | 170 | 2,719 |
| Complications costs | | | |
| Kidney replacement therapy[a] | 83,845 | 129,978 | -46,133 |
| ESKD (conservative therapy) | 2,698 | 2,908 | -211 |
| CVD complications | 12,245 | 11,266 | 979 |
| Anemia | 5,924 | 5,438 | 486 |
| Bone marrow disorder[b] | 13,690 | 12,114 | 1,576 |
| Acute kidney injury[c] | 2,059 | 2,048 | 11 |
| Infections | 11,005 | 9,944 | 1,061 |
| Cancers | 133 | 96 | 37 |
| Other[d] | 24,032 | 21,712 | 2,320 |
| Adverse events[e] | 310 | 104 | 206 |
| **Total** | **200,193** | **234,574** | **-34,380** |

[a]Includes kidney replacement therapy costs plus related complication costs

[b]Includes hyperphosphatemia, hypocalcemia, secondary hyperparathyroidism, fractures

[c]Includes acute kidney injury hospitalizations only

[d]Includes metabolic acidosis, hyperkalemia, and hyperuricemia/gout

[e]Includes lower limb (leg, toe, foot) amputations

Abbreviations: ESKD = end-stage kidney disease, CVD = cardiovascular disease, SoC = standard of care

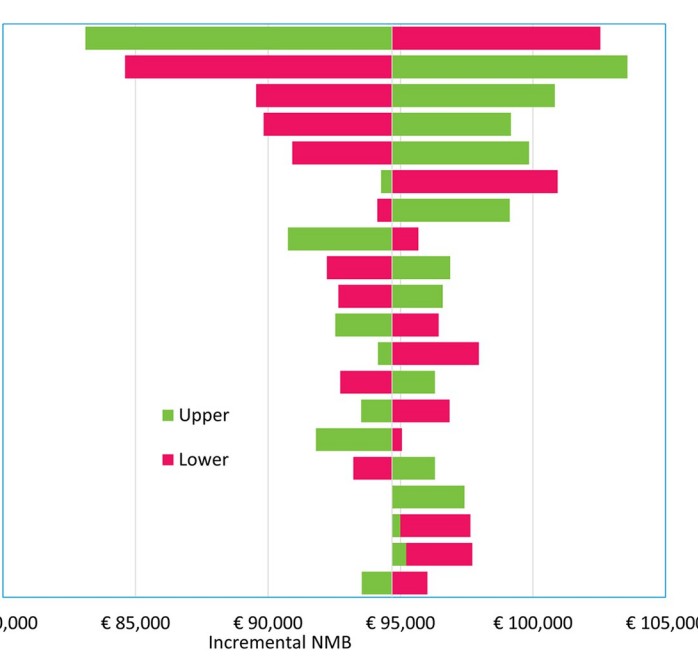

**Fig 2. Tornado diagram representing the results of the one-way sensitivity analysis.** Abbreviations: A = KDIGO uACR category (A1-A3), eGFR = estimated glomerular filtration rate, HD = hemodialysis, ESKD = end-stage kidney disease, G = KDIGO eGFR category, KDIGO = Kidney Disease: Improving Global Outcomes, NMB = net monetary benefit, RFP = risk factor progression, SoC = standard of care, uACR = urine albumin-creatinine ratio, EMPA = empagliflozin.

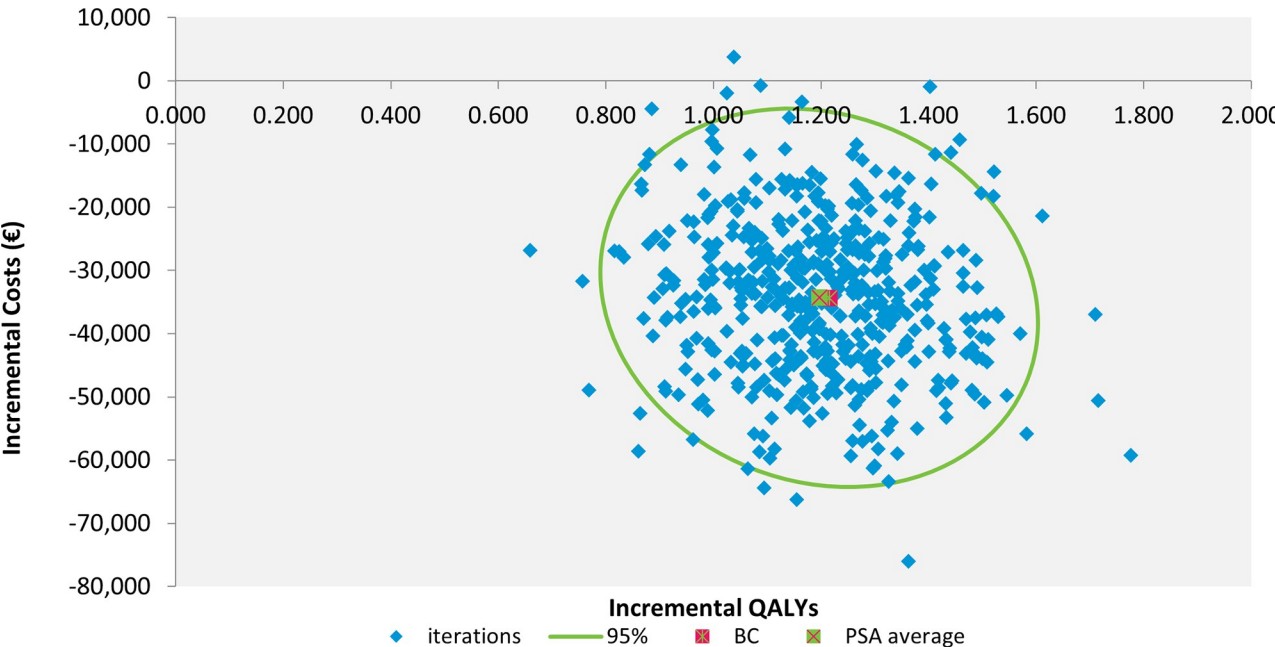

**Fig 3. Cost-effectiveness plane scatterplot showing the results of the probabilistic sensitivity analysis.** Abbreviations: QALY = quality-adjusted life year, 95% = 95% confidence interval, BC = base case, PSA = probabilistic sensitivity analysis.

### Scenario analyses

Table 3 presents the deterministic cost-effectiveness outcomes across various scenarios. All scenarios demonstrate dominant ICERs favoring empagliflozin + SoC over SoC alone, including the subgroup analyses exploring the difference between diabetic and non-diabetic patients (scenario 2 and 3), with most scenarios showing only a marginal effect on the outcomes.

### Discussion

We conducted a cost-effectiveness analysis based on the EMPA-KIDNEY trial, to estimate the potential health and economic outcomes of empagliflozin in patients with CKD in the Netherlands. Our analysis revealed that empagliflozin in addition to SoC is likely cost effective when compared to SoC alone, with a dominant ICER (cost saving while increasing the patient's quality of life). Empagliflozin remained highly cost-effective in both diabetic and non-diabetic patients.

In terms of health outcomes, empagliflozin plus SoC showed diminished incidence rates of all-cause hospitalizations and hospitalizations related to CKD, heart failure, and infections and a considerable increase in quality of life (1.22 QALYs gained per patient over lifetime) and life expectancy (1.31 years). The results are mainly driven by the reduction in CKD progression, especially the extended time to ESKD with an observed temporal difference of 10.79 years in the patients treated with empagliflozin and SoC versus 7.82 years in patients treated with SoC alone. Sensitivity analyses confirm the robustness of the results with an estimated probability of empagliflozin plus SoC being cost effective of 100% at a WTP-threshold of €50,000/QALY.

Adding empagliflozin to SoC results in a total cost saving of €34,380 per patient. A recent meta-analysis showed that in CKD the costs associated with renal events and heart failure are generally higher than those of atherosclerotic events [30]. Similarly, looking into the disintegrated costs in our results, it can be observed that the renal events are related with the highest

**Table 3. Deterministic cost-effectiveness analysis results of the scenario analysis.**

| Scenarios | | Discounted costs | Discounted LYs | Discounted QALYs | Incremental cost per LYs | Incremental cost per QALYs | NMB |
|---|---|---|---|---|---|---|---|
| **Base case** | | | | | | | |
| Empagliflozin plus SoC | | €200,193 | 11.06 | 9.01 | Dominant | Dominant | €95,183 |
| SoC | | €234,574 | 9.74 | 7.79 | | | |
| Incremental | | **-€34,380** | **1.31** | **1.22** | | | |
| **Scenario 1 –healthcare payer's perspective** | | | | | | | |
| Empagliflozin plus SoC | | €179,072 | 11.06 | 9.01 | Dominant | Dominant | €94,010 |
| SoC | | €212,279 | 9.74 | 7.79 | | | |
| Incremental | | **-€33,208** | **1.31** | **1.22** | | | |
| **Scenario 2 –EMPA-KIDNEY subgroup with diabetes** | | | | | | | |
| Empagliflozin plus SoC | | €193,259 | 10.05 | 8.15 | Dominant | Dominant | €85,972 |
| SoC | | €218,258 | 8.70 | 6.93 | | | |
| Incremental | | **-€24,999** | **1.35** | **1.22** | | | |
| **Scenario 3 –EMPA-KIDNEY subgroup without diabetes** | | | | | | | |
| Empagliflozin plus SoC | | €211,211 | 12.09 | 9.87 | Dominant | Dominant | €105,498 |
| SoC | | €256,582 | 10.83 | 8.67 | | | |
| Incremental | | **-€45,371** | **1.26** | **1.20** | | | |
| **Scenario 4 –patient baseline characteristics based on Dutch data from Sundström et al. [30]** | | | | | | | |
| Empagliflozin plus SoC | | €157,448 | 10.87 | 8.96 | Dominant | Dominant | €76,474 |
| SoC | | €177,494 | 9.63 | 7.83 | | | |
| Incremental | | **-€20,047** | **1.24** | **1.13** | | | |
| **Scenario 5 –health state utilities based on Jesky et al. [31]** | | | | | | | |
| Empagliflozin plus SoC | | €200,193 | 11.06 | 8.38 | Dominant | Dominant | €88,343 |
| SoC | | €234,574 | 9.74 | 7.30 | | | |
| Incremental | | **-€34,380** | **1.31** | **1.08** | | | |
| **Scenario 6 –using a 10 year time horizon** | | | | | | | |
| Empagliflozin plus SoC | | €119,934 | 7.41 | 6.11 | Dominant | Dominant | €58,288 |
| SoC | | €153,296 | 6.95 | 5.61 | | | |
| Incremental | | **-€33,363** | **0.46** | **0.50** | | | |
| **Scenario 7 –using a 20 year time horizon** | | | | | | | |
| Empagliflozin plus SoC | | €182,573 | 10.20 | 8.33 | Dominant | Dominant | €92,714 |
| SoC | | €222,439 | 9.11 | 7.28 | | | |
| Incremental | | **-€39,866** | **1.09** | **1.05** | | | |
| **Scenario 8 –including all the SoC costs for the comorbidities (as a weighted average)** | | | | | | | |
| Empagliflozin plus SoC | | €202,040 | 11.06 | 9.01 | Dominant | Dominant | €94,748 |
| SoC | | €235,986 | 9.74 | 7.79 | | | |
| Incremental | | **-€33,945** | **1.31** | **1.22** | | | |

Abbreviations: LY = life year, QALY = quality-adjusted life year, SoC = standard of care, NMB = net monetary benefit

costs. Especially the costs of KRT (dialysis and kidney transplantation) are an average cost of almost €130,000 per CKD patient over lifetime in the current SoC. This is where empagliflozin shows the most value in terms of cost savings, delivering savings of up to €46,000 per patient. These savings very well offset the higher treatment cost for empagliflozin. Some costs of other events are slightly higher in the empagliflozin plus SoC arm, likely caused by the better survival outcomes.

To our knowledge, this is the first cost-effectiveness analysis of a SGLT2 inhibitor in patients with CKD in the Netherlands. Other recent cost-effectiveness analyses on

empagliflozin have been conducted in the UK, Malaysia, Thailand and Vietnam which, like us, reported a dominant (UK, Malaysia, Thailand) or cost-effective (Vietnam) ICERs. Although input parameters were country specific, empagliflozin pricing was comparable to our study with £476,98 (€565,51) vs €511,31 for the UK while prices were somewhat lower in the Asian countries (ranging between €288.55 and €409.08 per year) [32, 33].

Furthermore, previous cost-effectiveness analyses in CKD patients in the Netherlands have been conducted for other SGLT2 inhibitors, canagliflozin and dapagliflozin. Comparable to our results, a dominant ICER was reported by the National Health Care Institute (*Zorginstituut Nederland)* for dapagliflozin plus SoC versus SoC alone based on the DAPA-CKD trial for reimbursement purposes in the Netherlands [18]. The DAPA-CKD trial was conducted to explore the effects of dapagliflozin which indicated similar improvements in reducing the CKD progression compared to empagliflozin in the EMPA-KIDNEY trial. In a recently published meta-analysis, the Nuffield Department of Population Health Renal Studies Group (2022) examined the effects of SGLT2 inhibitors on various outcome measures in patients with CKD with and without type 2 diabetes. The EMPA-KIDNEY and DAPA-CKD studies were included in the meta-analysis. They showed that the effect of dapagliflozin and empagliflozin on CKD progression is comparable, regardless of diabetes status [34]. The KDIGO concept guideline also makes no distinction between the two SGLT2 inhibitors [18]. Given that both empagliflozin and dapagliflozin have similar effectiveness and pricing in the Netherlands, the differences in cost-effectiveness between them are expected to be minimal. However, small differences between the trials may explain part of the small discrepancies in cost-effectiveness results.

Another cost-effectiveness analysis based on the DAPA-CKD trial from the payer's perspective was conducted for the UK, Germany and Spain showed ICERs of $8,280, $17,623 and $11,687 per QALY, with QALY gains of 0.82, 1.00 and 0.96 in the three countries respectively [35]. The difference compared with the results for the Dutch setting could include differences in modelling, input parameters, or pharmacoeconomic guidelines such as perspective and discounting. However, it would likely be a combination, as for example in our scenario from the payer's perspective the ICER remained still dominant. A cost-effectiveness study compared canagliflozin to dapagliflozin based on the DAPA-CKD and CREDENCE trials in patients with CKD and type 2 diabetes in Canada [36]. They concluded that adding either canagliflozin or dapagliflozin to SoC was dominant over SoC alone. However, canagliflozin is not reimbursed for CKD patients in the Netherlands. The cost effectiveness of dapagliflozin has also been evaluated in non-diabetic CKD patients based on the non-diabetic cohort of the DAPA-CKD trial in US setting, showing an ICER of $60,000 per QALY [37]. Overall, these analyses demonstrate that all SGTL2 inhibitors in addition to SoC are likely dominant over SoC alone.

For the purpose of this study, we conducted two subgroup analyses based on the diabetic and non-diabetic populations from the EMPA-KIDNEY trial. Although the results were relatively comparable between the groups and both resulted in dominant ICERs, the non-diabetic population seems to favor in terms of cost savings, while the diabetic population favors in terms of QALYs gained. This is consistent with findings from a single-center cross-sectional analysis, which suggests that SGTL2 inhibitors are often prescribed to higher-risk diabetic patients, potentially justifying their higher cost due to significant clinical benefits, particularly in cardiovascular and renal outcomes [38]. A Dutch study explored the cost effectiveness in CKD patients with diabetes, as it was shown that about 36% of the T2D population is associated with CKD [39]. This study investigated the effect of finerenone and showed potential for reducing the risk of CKD progression, saving costs for the society and increasing the QALYs compared to SoC. Further research is needed to compare these different drugs with each other for the Dutch societal setting.

A British research group performed a structured literature review to address the necessity of health state utilities for CKD and determinate the most appropriate inputs for economic models [40], including 4 Dutch studies. To best represent the population modelled, which was in our case the EMPA-KIDNEY population, we used the utilities directly measured in the trial and converted these into Dutch values. However, since there is uncertainty around the health state utilities for CKD in economic models, we conducted a scenario analysis based on health state utility values from Jesky et al. (also converted to Dutch values) [31]. This scenario showed a marginal effect on incremental QALYs, which was slightly less favorable with 1.08 instead of 1.22 QALYs gained.

There is evidence that studies using longer time-horizon tend to reflect more favorable ICERs [41]. In our study, from the two additional time horizons explored, indeed the longer one (20 years) showed more favorable outcomes compared to the scenario with a time horizon on 10 years. This estimation reflects the outcomes in the population from a mean age of 63.3 up to 83.3, the age category with the biggest costs according to van Oosten et al. who described this for the Netherlands [42].Therefore, the outcomes in the base case were less favorable than those in the scenario, even though the time horizon in the base case was even longer. Nevertheless, the time horizon does fit with the study's nature and ensures that all relevant consequences are being captured, which in our case was up to an age of 100 years.

## Limitations and strengths

This study has various limitations. Ideally, using Dutch population in the base case, instead of the trial population would have better reflect the situation of the Netherlands. One improvement point is for example the mean age of the population. In the EMPA-KIDNEY trial this was 63.8 years. A recent meta-analysis pooled the prevalence of CKD from populations of 11 countries with a mean age of 75-years-old [30]. Age-related differences in care and costs used in the Netherlands was made evident in a study based on health care claims [42]. Overall, the biggest costs among the CKD patients were in the age group of 65–74 years old population. This, together with other country specific patient characteristics, including disease severity may impact the disease progression and the need for treatments. Nevertheless, we have made efforts to explore this in scenario accounting for mean population age of 75 years and the proportion male of 54% (instead of 46% in base case) based on patient records data analyzed in the abovementioned meta-analysis. However, as the other patient characteristics did not match the inputs for our model, those kept the values from the EMPA-KIDNEY trial population, which does not comprehensively represent the pooled Dutch population. Therefore, the outcomes from these scenarios need to be interpreted with caution.

Furthermore, one may argue that the mortality sources for CKD patients should be based completely on a Dutch source rather than only those for the non-specific mortality. As no comprehensive information was further available to source the cardiovascular and renal disease related death, we used published sources for these inputs delivered from countries with comparable health and economic developments such as the UK. This is in line with the national guidelines for conducting economic evaluations [20]. Nevertheless, mortality is a parameter that can vary per country and model inputs should be updated when this data becomes available for the local situation.

Another limitation of the study may originate from the limitations of the EMPA-KIDNEY trial itself, which we used as a main source for our clinical inputs (such as baseline characteristics, utility values, and treatment effect) and the same treatment effect was applied across the model cycles. Yet, the disease progression based on eGFR evolution per KDIGO class was comparable to published results about natural CKD progression in the population [43].

Besides, patients with eGFR 45–60 ml/min/m2 without albuminuria were not included in the EMPA-Kidney trial, despite having CKD according to KDIGO criteria. According to the KDIGO guideline SGLT2 inhibitors are indicated in this population when they have heart failure as a comorbidity. Although this population was included in the model to properly model CKD progression, the treatment effect of empagliflozin was not included as there was no data from the EMPA-Kidney available for this group to support the effectiveness. Data on patients with eGFR between 45 and 60 ml/min, who meet KDIGO criteria but were excluded from the EMPA-kidney study, would provide valuable context and contribute to a more comprehensive understanding of empagliflozin's cost-effectiveness across different stages of CKD.

Finally, performing the analyses with real-world evidence may give different observations and results, as the actual population is not in a controlled condition like a randomized-controlled trial. One can imagine that RCTs provide limited insight into parameters such as adherence or safety in non-controlled conditions as reflected in the real-world, that might impact the efficacy of the treatment. Nevertheless, when introducing new treatments, real-world data is not always available and patient characteristics cannot be specified in detail, therefore initial use of RCT data is the way to start.

Nevertheless, we have performed a comprehensive analysis that includes risk factors and occurrence of events and complications using an individual simulation model. This is usually a great challenge considering the complexity of CKD management. The model was informed with patient-level data from the EMPA-KIDNEY trial where possible, which is generally more accurate than using average estimates from literature.

### Implications for use and future research

Since the expansion of the therapeutic field of SGLT2 inhibitors to the prevention of disease progression in CKD populations, there has been increasing interest in the cost effectiveness of these drugs. It is of interest to clinical decision makers to investigate the potential health economic benefits beyond the original indications in diabetic and heart failure patients. Given the high prevalence, and its related morbidity and mortality there is need for a systematic approach to fight CKD, including awareness, prevention, early detection and optimal care [11, 13]. Our results suggest empagliflozin would be a valuable and cost-effective addition to the current CKD therapy to slow CKD progression.

SGLT2 inhibitors may even have more value in combination with early detection of CKD based on eGFR and albuminuria. A recent Dutch study showed that screening for albuminuria to identify CKD in adult population to be cost effective [44]. The implementation of screening would lead to earlier detection and in combination with SGLT2 inhibitor treatment CKD progression, and its associated costs and health burden, could even be reduced further. This is important given the expected increase in CKD incidence and prevalence, and the increasing burden on the health care system in general.

The model developed for this study was created to address as many as possible risk factors, events and complications, it should be periodically updated to remain up to date accounting for new management alternatives. It would also be interesting to redo the analysis once (Dutch) real-world data using SGLT2 inhibitors in CKD becomes available.

### Conclusion

Using empagliflozin in addition to SoC in adult patients with CKD is likely to be cost saving compared to the current SoC in the Netherlands. Our study results support the implementation of empagliflozin in the current treatment management of CKD, irrespective of diabetes status and albuminuria.

## Supporting information

**S1 Appendix. Model structure.**
(DOCX)

**S1 Table. Baseline characteristics used in the model.**
(DOCX)

**S2 Table. Utility values used in the model.**
(DOCX)

**S3 Table. Cost inputs used in the model.**
(DOCX)

**S4 Table. Incidence of events per sub module for empagliflozin plus SoC versus SoC alone.**
(DOCX)

**S1 Fig. Cost-effectiveness acceptability curve presenting the results of the probabilistic sensitivity analysis.**
(DOCX)

## Author Contributions

**Conceptualization:** Maaike Weersma, Lisa de Jong.

**Data curation:** Bart Slob, Tanja Fens.

**Formal analysis:** Bart Slob, Tanja Fens, Lisa de Jong.

**Methodology:** Bart Slob, Tanja Fens, Maaike Weersma, Lisa de Jong.

**Supervision:** Maarten Postma, Cornelis Boersma, Lisa de Jong.

**Validation:** Maarten Postma, Cornelis Boersma.

**Visualization:** Bart Slob, Tanja Fens.

**Writing – original draft:** Bart Slob, Tanja Fens.

**Writing – review & editing:** Maaike Weersma, Maarten Postma, Cornelis Boersma, Lisa de Jong.

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
