## [Decision Letter · Decision Letter 0]

19 Aug 2024

PONE-D-24-30732Cost Effectiveness of Empagliflozin in Adult Patients with Chronic Kidney Disease in the NetherlandsPLOS ONE

Dear Dr. Slob,

Thank you for submitting your manuscript to PLOS ONE. After careful consideration, we feel that it has merit but does not fully meet PLOS ONE’s publication criteria as it currently stands. Therefore, we invite you to submit a revised version of the manuscript that addresses the points raised during the review process.

We look forward to receiving your revised manuscript.

Kind regards,

Larry Allan Weinrauch, MD

Academic Editor

PLOS ONE

Journal Requirements:

"This study was funded by Boehringer Ingelheim Netherlands."

"CB and MP receive grants and honoraria from various pharmaceutical companies, including Boehringer Ingelheim. They are both shareholders of Health-Ecore, the Netherlands. TF, BS and LJ are employed as consultants at Health-Ecore, which received a consultancy fee for the conduct of this study. MW is an employee at Boehringer Ingelheim Netherlands, the funder of this study. The EMPA-KIDNEY trial was initiated, designed, and conducted by the University of Oxford in collaboration with a Steering Committee of experts and Boehringer Ingelheim. The presented analyses were initiated and conducted by Boehringer Ingelheim independently from the EMPA-KIDNEY Collaborative Group. The authors meet criteria for authorship as recommended by the International Committee of Medical Journal Editors (ICMJE). The authors did not receive payment related to the development of this manuscript. Boehringer Ingelheim was given the opportunity to review the manuscript for medical and scientific accuracy as well as intellectual property considerations."

We note that one or more of the authors are employed by a commercial company: Health-Ecore, Boehringer Ingelheim Netherlands

2) Please also provide an updated Competing Interests Statement declaring this commercial affiliation along with any other relevant declarations relating to employment, consultancy, patents, products in development, or marketed products, etc.  

Within your Competing Interests Statement, please confirm that this commercial affiliation does not alter your adherence to all PLOS ONE policies on sharing data and materials by including the following statement: ""This does not alter our adherence to  PLOS ONE policies on sharing data and materials.” (as detailed online in our guide for authors http://journals.plos.org/plosone/s/competing-interests) . If this adherence statement is not accurate and  there are restrictions on sharing of data and/or materials, please state these. Please note that we cannot proceed with consideration of your article until this information has been declared.

**Additional Editor Comments:**

*The analysis of the Cost Effectiveness of Empagliflozin in Adult Patients with Chronic Kidney Disease in the Netherlands purports to suggest that the addition of empagliflozin to standard of care medications might be cost saving. Several points have been brought up by the reviewers that do need to be addressed. One, of course is the influence of industry upon the presentation. Another is on the price of empagliflozin and whether it is similar in other countries (particularly with respect to those outside of the EU). Yet another, and perhaps even more important is that not all CKD is the same. A large proportion of CKD involves patients with diabetes. It would be helpful to break down the differences in the value of adding empagliflozin to standard of care with respect to nondiabetic vs diabetic patients, or perhaps those who are underweight vs. those who are overweight. This would give us additional information (e.g J Diabetes Complications 2021 Feb;35(2):107761)*

Please respond to the requests made by the reviewers as well.

Reviewers' comments:

Reviewer's Responses to Questions

**Comments to the Author**

1. Is the manuscript technically sound, and do the data support the conclusions?

Reviewer #1: Yes

Reviewer #2: Yes

2. Has the statistical analysis been performed appropriately and rigorously? 

Reviewer #1: Yes

Reviewer #2: Yes

3. Have the authors made all data underlying the findings in their manuscript fully available?

Reviewer #1: No

Reviewer #2: Yes

4. Is the manuscript presented in an intelligible fashion and written in standard English?

Reviewer #1: Yes

Reviewer #2: Yes

5. Review Comments to the Author

Reviewer #1: This study provides valuable insights into the cost-effectiveness of empagliflozin for patients with chronic kidney disease (CKD) in the Netherlands. It is commendable that the researchers adhered to the KDIGO guidelines for CKD definition, which include both GFR levels and for eGFR >60 mll/min also the evidence of kidney damage, such as albuminuria or structural abnormalities.

The Empa Kidney study inclusion criteria, focusing on patients with eGFR less than 45 ml/min/1.73 m2 or those with eGFR between 45 and 90 ml/min/1.73 m2 and albuminuria, ensured that all participants met the KDIGO criteria. The current study presentation, particularly Figure 1, does not adequately address this aspect.

Additionally, it would be insightful to know if patients with eGFR greater than 60 ml/min without albuminuria or other signs of kidney damage, who would not meet the KDIGO criteria for CKD, were included. Furthermore, data on patients with eGFR between 45 and 60 ml/min, who meet KDIGO criteria but were excluded from the Empa kidney study, would provide valuable context and contribute to a more comprehensive understanding of empagliflozin's cost-effectiveness across different stages of CKD.

Reviewer #2: Slob et al present a detailed cost effectiveness analysis of empagliflozin in the Netherlands comparing standard of care for chronic kidney disease plus empagliflozin (empa) and standard of care alone (SoC0. Using a Markov model they show that the cost effectiveness of SoC+empag versus SoC alone is a savings of 34,381 Euros. They show that SoC+empa adds an additional 1.22 life years compared to SoC alone. The authors present data on CKD in the Netherlands that shows prevalence around 12% but only 5.1% of the Dutch population is diagnosed. I have the following questions.

1. Why was only empagliflozin studied and not the other major SGLT2 inhibitors (dapagliflozin and canagliflozin)? Are there cost analyses for them and how does the cost effectiveness of empagliflozin compare to the cost effectiveness of the others?

2. What is the standard of care for CKD in the Netherland and does it vary from co-morbidity to co-morbidity (diabetes, heart failure, coronary artery disease. How does the analysis take account of the fact that standard cof care might differ with underlying co-morbidities. The authors provide the details from KDIGO (lines 82-93, but should include regimens specific to the Netherlands.

3. How does Cost-effectiveness of empagliflozin and SoC differ across the European Union? What was the need to calculate cost effectiveness for empagliflozin specifically for the Netherlands?

6. PLOS authors have the option to publish the peer review history of their article (what does this mean?). If published, this will include your full peer review and any attached files.

Reviewer #1: **Yes: **Bijan Roshan

Reviewer #2: **Yes: **George Bayliss, MD

---

## [Author Response · Author response to Decision Letter 0]

5 Nov 2024

Dear editors,

Dear reviewers,

Thank you for your time and valuable comments regarding our work. Please see the attached Word files for the specific responses and also for the updated funding and conflict of interest statements. Please find our response in italic after each of your comments and see the track changes in the updated manuscript. 

On behalf of all authors,

Kind regards,

Bart Slob

---

## [Decision Letter · Decision Letter 1]

27 Nov 2024

Cost effectiveness of empagliflozin in adult patients with chronic kidney disease in the Netherlands

PONE-D-24-30732R1

Dear Dr. Slob

We’re pleased to inform you that your manuscript has been judged scientifically suitable for publication and will be formally accepted for publication once it meets all outstanding technical requirements.

Kind regards,

Larry Allan Weinrauch, MD

Academic Editor

PLOS ONE

Additional Editor Comments (optional):

My apologies for the long delay in accepting this work. The concerns of the reviewers and the editor have been satisfied in the revision

Reviewers' comments:

Reviewer's Responses to Questions

**Comments to the Author**

1. If the authors have adequately addressed your comments raised in a previous round of review and you feel that this manuscript is now acceptable for publication, you may indicate that here to bypass the “Comments to the Author” section, enter your conflict of interest statement in the “Confidential to Editor” section, and submit your "Accept" recommendation.

Reviewer #1: All comments have been addressed

Reviewer #2: All comments have been addressed

2. Is the manuscript technically sound, and do the data support the conclusions?

Reviewer #1: Yes

Reviewer #2: Yes

3. Has the statistical analysis been performed appropriately and rigorously? 

Reviewer #1: (No Response)

Reviewer #2: Yes

4. Have the authors made all data underlying the findings in their manuscript fully available?

Reviewer #1: Yes

Reviewer #2: Yes

5. Is the manuscript presented in an intelligible fashion and written in standard English?

Reviewer #1: Yes

Reviewer #2: Yes

6. Review Comments to the Author

Reviewer #1: (No Response)

Reviewer #2: The authors have addressed my concerns. I think they addressed the commercial interest of the pharmaceutical company, provided information about other SGLT2 inhibitors in the context of the Dutch healthcare system and noted the need for individual drug analysis

7. PLOS authors have the option to publish the peer review history of their article (what does this mean?). If published, this will include your full peer review and any attached files.

Reviewer #1: **Yes: **Bijan Roshan

Reviewer #2: **Yes: **George Bayliss, MD, FACP, FASN

---

## [Editor Report · Acceptance letter]

29 Nov 2024

PONE-D-24-30732R1 

PLOS ONE

Dear Dr. Slob, 

I'm pleased to inform you that your manuscript has been deemed suitable for publication in PLOS ONE. Congratulations! Your manuscript is now being handed over to our production team.

Kind regards, 

on behalf of

Dr. Larry Allan Weinrauch 

Academic Editor

PLOS ONE